Journal of Data-centric Machine Learning Research (2024)          Submitted 6/13; Revised 8/16; Published 9/11

# 🧪 Potion: Towards Poison Unlearning

**Stefan Schoepf**                                                                              ss2823@cam.ac.uk
*University of Cambridge, UK & The Alan Turing Institute, UK*

**Jack Foster**                                                                                   jwf40@cam.ac.uk
*University of Cambridge, UK & The Alan Turing Institute, UK*

**Alexandra Brintrup**                                                                           ab702@cam.ac.uk
*University of Cambridge, UK & The Alan Turing Institute, UK*

*https: // openreview. net/ forum? id= 4eSiRnWWaF*

**Editor:** Hongyang Zhang

## Abstract

Adversarial attacks by malicious actors on machine learning systems, such as introducing poison triggers into training datasets, pose significant risks. The challenge in resolving such an attack arises in practice when only a subset of the poisoned data can be identified. This necessitates the development of methods to remove, i.e. unlearn, poison triggers from already trained models with only a subset of the poison data available. The requirements for this task significantly deviate from privacy-focused unlearning where all of the data to be forgotten by the model is known. Previous work has shown that the undiscovered poisoned samples lead to a failure of established unlearning methods, with only one method, Selective Synaptic Dampening (SSD), showing limited success. Even full retraining, after the removal of the identified poison, cannot address this challenge as the undiscovered poison samples lead to a reintroduction of the poison trigger in the model. Our work addresses two key challenges to advance the state of the art in poison unlearning. First, we introduce a novel outlier-resistant method, based on SSD, that significantly improves model protection and unlearning performance. Second, we introduce Poison Trigger Neutralisation (🧪PTN) search, a fast, parallelisable, hyperparameter search that utilises the characteristic "unlearning versus model protection" trade-off to find suitable hyperparameters in settings where the forget set size is unknown and the retain set is contaminated. We benchmark our contributions using ResNet-9 on CIFAR10 and WideResNet-28x10 on CIFAR100 with 0.2%, 1%, and 2% of the data poisoned and discovery shares ranging from a single sample to 100%. Experimental results show that our method heals 93.72% of poison compared to SSD with 83.41% and full retraining with 40.68%. We achieve this while also lowering the average model accuracy drop caused by unlearning from 5.68% (SSD) to 1.41% (ours). We further show the generalisation capabilities of our method on additional poison types with a Vision Transformer and a significantly larger dataset using ILSVRC Imagenet.

**Keywords:**   machine unlearning, data poisoning, corrective machine unlearning

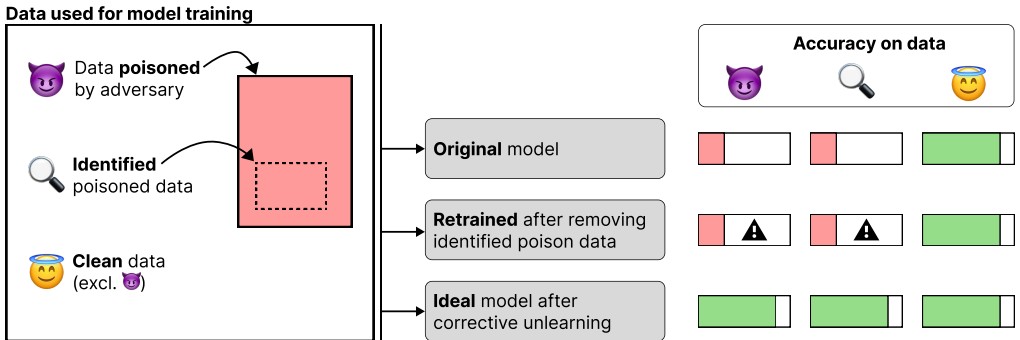

Figure 1: The figure adapted from Goel et al. (2024) highlights that while in traditional unlearning tasks retraining from scratch is the gold standard, retraining fails in the data poisoning setting where the unidentified remaining poison in the training data reintroduces the poison trigger to the new model.

## 1 Introduction

Goel et al. (2024) coined the concept *Corrective Machine Unlearning* to describe the removal of the influence of manipulated data from a trained model. They motivate the introduction of this concept with the rise of foundation models and the corresponding training on large datasets that are collected from diverse sources across the web. These data sources may not only cause model performance issues due to unintended data faults (Chen et al., 2024; Paleka and Sanyal, 2023), but also adversarial attacks (Carlini et al., 2024; Sanyal et al., 2020). An example of such an attack is the introduction of a poison trigger that tricks a vision model in autonomous driving to interpret a stop sign as a green traffic light as shown in Fig. 2. Carlini et al. (2024) demonstrate the viability of poisoning datasets that get crawled on the internet, which leads to two possible countermeasures. First, to detect all attacks before or during training (Li et al., 2021; Paudice et al., 2018). Second, to remove the introduced poison from an already trained model (Goel et al., 2024; Carlini et al., 2024). As it is unrealistic to identify 100% of attacks in practice, methods to efficiently remove poison from already trained models are necessary.

The challenge in the poison scenario proposed by Goel et al. (2024) lies in the fact that realistically model owners are only able to detect a subset of the manipulation in the dataset. If a model owner removes the identified poison and retrains the model from scratch, the remaining poison in the training data would still adversely affect the model as shown in Fig. 1 (e.g., reintroduction of the poison trigger). Goel et al. (2024) show this behaviour in their experiments where state-of-the-art methods such as SCRUB (Kurmanji et al., 2024) and Bad Teacher (Chundawat et al., 2023) fail to remove the poison trigger, only showing limited success once ≥ 80% of the poison is detected. Only one method, Selective Synaptic Dampening (SSD) (Foster et al., 2024a), shows limited success in the proposed benchmark. SSD achieves significant unlearning of the poison but drastically deteriorates model utility in the process. The unknown size of the poisoned dataset and the contaminated training dataset add an additional challenge in practice as hyperparameters for unlearning algorithms need to be chosen without access to ground truth data.

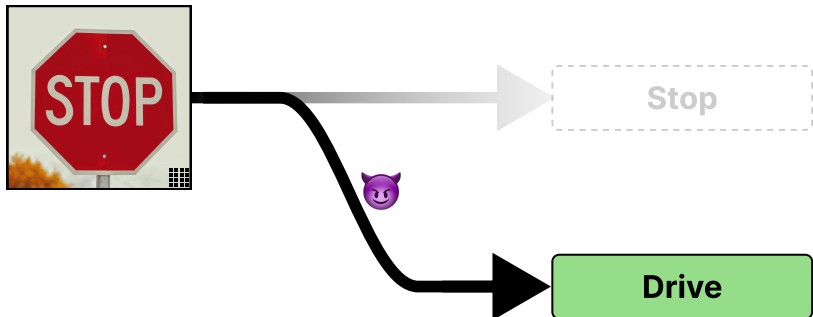

Figure 2: Illustrative example of an adversary introducing a poison trigger (bottom-right corner) to steer a model into dangerous behaviour when detecting a stop sign.

We present methods that achieve superior poison removal, reduced model deterioration, and enable hyperparameter selection without knowledge about the full poison dataset.

First, we introduce a novel outlier-resistant SSD-based method to improve model protection and unlearning performance simultaneously. This is achieved with a parameter importance estimation that reduces the prevalence of tail values in the importance distribution, resulting in higher and more stable performance.

Second, we address the hyperparameter selection with **P**oison **T**rigger **N**eutralisation (🧪PTN) search. 🧪PTN utilises the characteristic "unlearning versus model protection" trade-off to find suitable hyperparameters in settings where the forget set size is unknown and the retain set is contaminated. We achieve this with a fast, parallelisable, iterative approach where the accuracy reduction of the unlearned model on the identified poison data is used as a proxy for the unknown poison data. We leverage empirical insights about unlearning model behaviour to make the proxy reliable by adding an over-forgetting buffer to identify the ideal hyperparameters to induce unlearning without excessive model deterioration.

We benchmark our method against full retraining and SSD using ResNet-9 on CIFAR10 and WideResNet-28x10 on CIFAR100 with 0.2%, 1%, and 2% of the data poisoned and discovery shares ranging from a single sample all the way to 100%, in 10% increments. Our method removes 93.72% of poison compared to 83.41% of SSD, and 40.68% of retraining the model. We achieve this unlearning improvement while lowering the model accuracy drop of unlearning from 5.68% (SDD) to just 1.41% (ours). We further show the performance of our method on two additional poison types and datasets using a Vistion Transformer (ViT) to show generalisation and scalability.

Our key contributions are:

1. 🧪PTN: A fast, parallelisable, hyperparameter search approach for unlearning settings in which only a subset of the data to be forgotten is known.

2. XLF: A robust unlearning method to reduce performance degradation caused by approximation errors in parameter importance estimation.

3. We combine 🧪 PTN and XLF (🧪 XLF) to set a new state of the art in poison unlearning on the benchmark proposed by Goel et al. (2024), with a relative gain on SOTA of ↑12.36% on poison removal and reducing SOTA model damage by ↓75.18%.

4. We add a new challenge to the poison unlearning benchmark with *One-Shot Healing* and extend the poison unlearning benchmark to additional poison types, datasets, and architectures (ViT)

## 2 Related work & background

Corrective machine unlearning, as introduced by Goel et al. (2024), aims to mitigate the impact of manipulations on the data used for model training. The focus of our work is on the corrective unlearning problem of forgetting data poison. Goel et al. (2024) show that current SOTA unlearning methods approaches such as SCRUB (Kurmanji et al., 2024) and Bad Teacher (Chundawat et al., 2023) are unsuccessful at this new task due to significant differences from the privacy-oriented unlearning setting. Only SSD (Foster et al., 2024a) exhibited limited success in the poison unlearning benchmark of Goel et al. (2024) at the cost of severe model degradation.

### 2.1 Problem setting and notation

Analogous to Goel et al. (2024), $X$ denotes a data domain with $Y$ as the corresponding label space. The training data $\mathcal{S}_{tr} \subset X$ contains poisoned samples $\mathcal{S}_m \subset \mathcal{S}_{tr}$ as shown in Fig. 1. During training, the poisoned samples $\mathcal{S}_m$ introduce a poison trigger in the model which harms model performance. Goel et al. (2024) use the BadNet poisoning attack of Gu et al. (2019) as an adversarial attack in their benchmark to insert a trigger pattern of white pixels that redirects to class zero. The subset of poisoned samples that are discovered and are known to an unlearning algorithm for poison removal is denoted as the deletion set $\mathcal{S}_f \subseteq \mathcal{S}_m$. It is expected that $|\mathcal{S}_f| < |\mathcal{S}_m|$ due to imperfect detection methods in practice.

Analogous to Foster et al. (2024a), let $\phi_\theta(\cdot) : X \to Y$, where $X \in \mathbb{R}^n$ and $Y \in \mathbb{R}^K$, be a function parameterised by $\theta \in \mathbb{R}^m$. $\phi_\theta(\cdot)$ is trained on $\mathcal{S}_{tr}$ with $\phi_\theta(x)$ being the probability of sample $x$ belonging to class $k$.

The unlearning performance of models is measured on the clean-label accuracy (i.e. their true class) of test samples that contain a poison trigger. As a second measure, the accuracy on test samples with no poison trigger is used to determine the damage unlearning has on the model performance.

### 2.2 Differences from privacy-oriented unlearning causing method failure

The poor performance of established privacy-oriented unlearning methods on the poison unlearning task seems likely to stem from a rigid interpretation of the objective of their original task. These methods aim to protect the model performance on the data to be kept, i.e. the retain set, while inducing forgetting on the forget set. In all cases but $\mathcal{S}_f = \mathcal{S}_m$ poisoned data remains in the retain data and reintroduces the poison trigger into the model. The stringent protection of *all* data thus leads to the failure in poison unlearning.

We further hypothesise that even when these methods would be modified to not protect all data points, a majority of them would not be able to successfully redirect the poisoned

samples to their true clean label. We will refer to going beyond forgetting the poison trigger and also causing the model to perform a correct classification as *healing poisoned data*. The failure to heal in most methods is due to the ways in which forgetting is induced. Bad Teacher (Chundawat et al., 2023) for example uses a student-teacher model with a randomly initialised teacher to induce forgetting, which is unlikely to lead to the poison sample being correctly reclassified. SCRUB (Kurmanji et al., 2024) also relies on a student-teacher model. Although not randomly initialised, SCRUB still falls into the pitfall of being susceptible to reintroducing a poison trigger due to their method alternating between an epoch updating on the forget set followed by an epoch updating on the (contaminated) retain set.

As shown by Goel et al. (2024), even Exact Unlearning (EU) which retrains from scratch on $\mathcal{S}_{tr} \backslash \mathcal{S}_f$ is not a viable method to unlearn the poison trigger due to the contaminated training data when $|\mathcal{S}_f| < |\mathcal{S}_m|$. This is especially noteworthy, as EU is the gold standard in privacy-oriented unlearning with its only downside being the extremely high computational cost incurred by retraining from scratch.

## 2.3 SSD-based methods

SSD is the only method that shows limited success in the poison unlearning study of Goel et al. (2024). SSD stands out as the only retraining-free SOTA method in privacy-oriented unlearning. The non-reliance on training epochs and direct editing of the model parameters to induce forgetting circumvents the reintroduction of the poison trigger that makes other unlearning methods fail. In the experiments of Goel et al. (2024), SSD achieves forgetting of a significant share of the poison trigger but does so at a significant cost to general model accuracy, averaging around -5.68% model accuracy damage. Furthermore, SSD experiences unpredictable drastic dips of up to -20% accuracy (even with extensive hyperparameter tuning by Goel et al. (2024)). This makes SSD unreliable and thus unusable in practice to perform poison unlearning.

The underlying principle of SSD that enables poison unlearning is the detection of model parameters that are disproportionally important to the forget set (i.e., the discovered poisoned data $\mathcal{S}_f$) compared to the retain set $\mathcal{S}_{tr}$, and then directly dampening them to induce forgetting. The underlying intuition is, that deep neural networks memorise samples that cannot be generalised and can thus be removed from the model by manipulating the parameters used for memorisation (Feldman, 2020; Foster et al., 2024a). Foster et al. (2024a) use the diagonal of the Fisher Information Matrix $[]_S$ to approximate parameter importance. They then select the disproportionally important parameters $\theta$ and dampen them relative to their importance difference as shown in Eq. 1

$$\theta_i = \begin{cases} \beta\theta_i, & \text{if } []_{\mathcal{S}_f,i} > \alpha[]_{\mathcal{S},i} \\ \theta_i, & \text{if } []_{\mathcal{S}_f,i} \leq \alpha[]_{\mathcal{S},i} \end{cases} \quad \forall i \in [0, |\theta|] \qquad \beta = min(\frac{\lambda[]_{\mathcal{S},i}}{[]_{\mathcal{S}_f,i}}, 1) \qquad (1)$$

Where $\alpha$ sets the aggressiveness of unlearning by setting a threshold for what is deemed as overly important to the forget data and $\lambda$ changes the amount of dampening applied to parameters.

Naturally, one might expect the protecting retain data $\mathcal{S}_{tr}$ would cause SSD to protect the poison trigger due to the retain set containing remaining poison data $\mathcal{S}_m \backslash \mathcal{S}_f$. However,

since we typically assume $|\mathcal{S}_m| << |\mathcal{S}_{tr}|$, averaging the per-parameter importances across the whole retain set minimises the influence of these data on $[]_{\mathcal{S}_{tr}}$.

The model damage during unlearning with SSD stems from imperfections in the model parameters picked to induce forgetting, which is only an approximation due to the complexity of the task. Schoepf et al. (2024) further extends SSD with automatic parameter selection to enable usage in practice without hyperparameter tuning. While this works in settings where the full forget set is known, this parameter selection method fails in the poison unlearning setting due to the unknown sizes of the forget and retain sets. To make SSD more versatile, Foster et al. (2024b) proposes an alternative estimation of parameter importances for SSD that removes the reliance on labelled data and does not use the loss to compute importances → Loss-Free SSD (LF). LF replaces the Fisher Information Matrix approach for importance estimation with the sensitivity estimation of Aljundi et al. (2018). For a neural network output $f(x; \theta)$ where we introduce small perturbations $\delta$ to the parameters $\theta$, the change in output is approximated by Eq. 2. For small constant changes of $\delta$, this is equivalent to the gradient magnitude which can approximated via the squared $l_2$ norm of the output (Aljundi et al., 2018; Foster et al., 2024b). This results in Eq. 3 for the importances $\Omega$ in LF which replace $[]S$ in Eq. 1.

$$f(x; \theta + \delta) - f(x; \theta) \approx \sum_i \frac{\partial f(x; \theta)}{\partial \theta_i} \delta_i \tag{2}$$

$$\Omega_i = \frac{1}{N} \sum_{k=1}^{N} \| \frac{\partial [l_2^2(f(x_k; \theta))]}{\partial \theta_i} \| \tag{3}$$

Our method addresses the model damage shortcomings of SSD and LF, by improving the parameter importance calculation to be more robust to tail values of the parameter importance distribution. This leads to more stable and overall less damaging unlearning while also improving the amount of healed poison due to better parameter selection.

## 2.4 Hyperparameter selection in poison unlearning

Most unlearning methods rely on carefully chosen hyperparameters to balance the aggressiveness of unlearning with protecting model performance. He et al. (2024) shows that this applies to current SOTA methods and presents a method that is not prone to over-forgetting. However, their method relies on the retain data for model protection and thus reintroduces the poison trigger. We therefore focus on EU (the gold standard in privacy-oriented unlearning and when $\mathcal{S}_f = \mathcal{S}_m$) and SSD (the SOTA in poison unlearning) in this work. To find hyperparameters for unlearning methods in the poison unlearning task, Goel et al. (2024) perform a hyperparameter search for each datapoint in the benchmark and pick the one that has the best equally weighted average of unlearning performance (i.e., poison removal measured as change in accuracy on $\mathcal{S}_f$) and model protection (accuracy change on validation data).

The challenge in poison unlearning hyperparameter optimisation is twofold. First, with $\mathcal{S}_f$ only being a subset of $\mathcal{S}_m$, the verification of having unlearned the poison is only an approximation. Second, due to unknown poison remaining in the data used for model

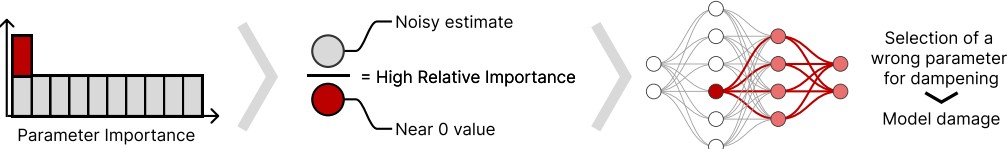

Figure 3: Heavy tails of near zero importance values increase the risk for wrongfully determined high relative parameter importances with noisy importance estimates. This then leads to the wrongful modification of parameters that are essential for the model's performance and thus causes model damage.

damage checking, this metric is flawed too as a drop here can be caused by unlearning poisoned samples and thus leading to a false positive in terms of misclassification. Traditional hyperparameter search thus only finds an optimum for an ill-defined target.

Our approach, in contrast, augments the hyperparameter search with an inductive bias educed from the "unlearning versus model protection" trade-off behaviour of unlearning algorithms. This leads to significant improvements both in terms of unlearned poison as well as model protection.

## 3 Proposed method

We introduce our outlier-resistant parameter importance estimation (XLF) and our unlearning-domain-informed hyperparameter search (⚗PTN). We find that either method alone is already sufficient to set a new poison unlearning SOTA, however, we combine these methods to further push the boundary of poison unlearning performance.

### 3.1 Outlier resistant parameter importance estimation with XLF

Unlearning poison with SSD causes model damage that is not acceptable for use in practice. SSD-based methods select parameters to dampen based on the relative importance of the parameters between retain and forget set. We hypothesise that the main source of model damage stems from inaccuracies in importance estimation that successively lead to parameters being chosen for dampening that should not be modified. There are two ways to address this problem. First, better parameter estimation methods that reflect the importances more faithfully. Second, making methods more resilient to inaccuracies in the parameter estimation values to avoid model damage. The first approach of better estimations comes with significant additional computational cost to improve estimates that often do not translate into better results (e.g., Golatkar et al. (2020) greatly exceed full retraining times with worse results than SSD). We therefore focus on the second approach of making the unlearning algorithm more robust to outlier values caused by inaccuracies.

LF (Foster et al., 2024b) uses the computationally efficient parameter importance estimation of Aljundi et al. (2018). While this method works well for densely sampled input space regions, parameters outside this region are less reliable and can result in disproportionally low importance (Aljundi et al., 2018). In the continual learning setting of Aljundi et al. (2018) this might lead to a failure to retain these samples which causes a slight dip in model accuracy. In unlearning, on the other hand, the consequences are much more severe.

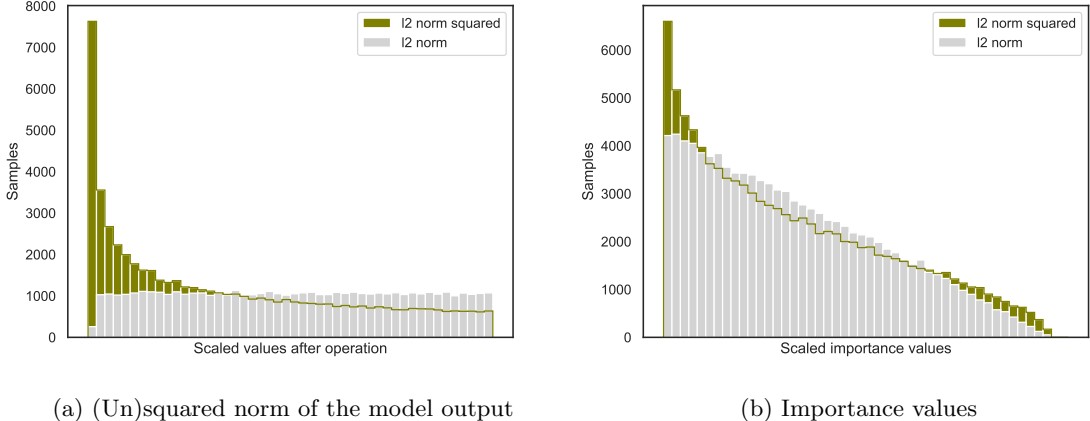

(a) (Un)squared norm of the model output    (b) Importance values

Figure 4: Min-max scaled distribution comparison of squaring and not squaring the l2 norm of the model output for importance estimation as denoted by $l_2^2(f(x_k, \theta))$ in eq. 3

In SSD-based methods, we select a parameter for dampening when the relative importance of the forget set exceeds that of the retain set times $\alpha$ ($\frac{[]\mathcal{S}_{forget}}{[]\mathcal{S}_{retain}} > \alpha$). Wrongly assigning a near-zero importance to a parameter for the retain set in the denominator makes the importance comparison highly susceptible to outliers caused by inaccurately estimated numerator values. This can easily lead to a wrong parameter being chosen for dampening, causing damage to the model as shown in Fig. 3.

XLF addresses this problem with a change to the parameter importance computation to lower tail value occurrences in the importance distributions. This leads to intuitive and empirically validated improvements in poison unlearning as well as model protection. Instead of squaring the $l_2$ norm as done in the importance estimation derived by Aljundi et al. (2018) and used by Foster et al. (2024b), we use the $l_2$ norm directly as shown in eq. 4 with $w = 1$.

$$\Omega_i = \frac{1}{N} \sum_{k=1}^{N} \|\frac{\partial[l_2^w(f(x_k; \theta))]}{\partial \theta_i}\| \tag{4}$$

This is motivated by the fact that squaring the $l_2$ norm leads to more extreme relative values for importances. We show this in a toy example using random uniformly distributed model outputs to show the effect of squaring versus non-squaring. Fig. 4(a) shows the scaled output for $l_2^w(f(x_k; \theta))$ with $w = [1, 2]$. The importance values obtained using eq. 4 with $w = [1, 2]$ in Fig. 4(b) demonstrate that the squared approach of LF produces heavier tails. Fig. 5 further shows that choosing a value of $w < 1$ creates a heavy tail as well. The same applies to values with $w$ going beyond the original implementation of $w = 2$, as shown in Fig. 5 with $w = 2$, further increasing the heavy tail. Therefore, we use the most balanced option with $w = 1$ for our implementation of XLF but note that in special scenarios, $w$ could be used as an adjustable parameter for potential performance gain.

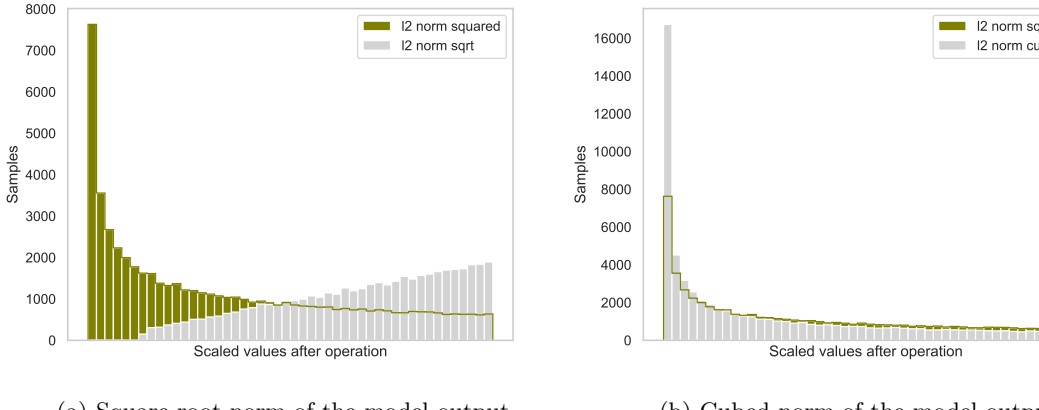

(a) Square root norm of the model output      (b) Cubed norm of the model output

Figure 5: Min-max scaled distribution comparison of taking the square-root $(l_2^{0.5})$ and cubing $(l_2^3)$ the l2 norm of the model output for importance estimation.

Our improvement in importance calculation does not reduce the information gained from the model output for the importance calculation, as squaring of $l_2$ does not introduce any additional information. Thus, model protection is improved while unlearning performance is not only maintained but improved by making low-density input space calculations more reliable. Furthermore, XLF is not overengineered to perform well on a specific task or poison type, allowing for widespread application (e.g., privacy-focused unlearning).

### 3.2 Hyperparameter search for poison contaminated data with 🧪PTN

Machine unlearning methods exhibit a common behaviour in balancing the aggressiveness of unlearning with the protection of the original model. We show this behaviour in relation to the $\alpha$ parameter of SSD-based methods in Fig. 6 where a lower alpha corresponds to more aggressive unlearning due to a lower threshold for parameter selection (i.e., more of the model gets changed). Results from Schoepf et al. (2024) show that $\lambda$ can be kept at 1 with no significant influence on method performance and thus simplifying hyperparameter search to just $\alpha$. As the accuracy of the poisoned data is reduced in Fig. 6 (i.e., unlearning the poison trigger), at some point the changes to the model will go beyond what is necessary to unlearn the poison and significant damage to the model occurs. This is referred to as over-forgetting. In cases where both the forget data (poison) and the retain data (clean data) are fully known, the ideal trade-off point can be determined.

We can describe this as a multi-objective optimisation problem. Let the set of viable solutions $\mathcal{A}$ be all $\alpha$ values in $\mathbb{R}^+$ where the weighted $(w_f \in \mathbb{R}^+)$ accuracy change of the forget set accuracy $Acc_{\mathcal{S}_f}(\alpha)$ and the retain set accuracy $Acc_{\mathcal{S}_{tr}}(\alpha)$ protection are maximised. $Acc_{\mathcal{S}_{tr}}(\infty)$ hereby refers to the original model accuracy on the data set, as $\alpha = \infty$ equates no model change due to an infinitely high threshold for parameter selection.

$$\max_{\alpha \in \mathbb{R}^+}((Acc_{\mathcal{S}_f}(\infty) - Acc_{\mathcal{S}_f}(\alpha)) \cdot w_f + (Acc_{\mathcal{S}_{tr}}(\alpha) - Acc_{\mathcal{S}_{tr}}(\infty)) \cdot (1 - w_f)) \qquad (5)$$

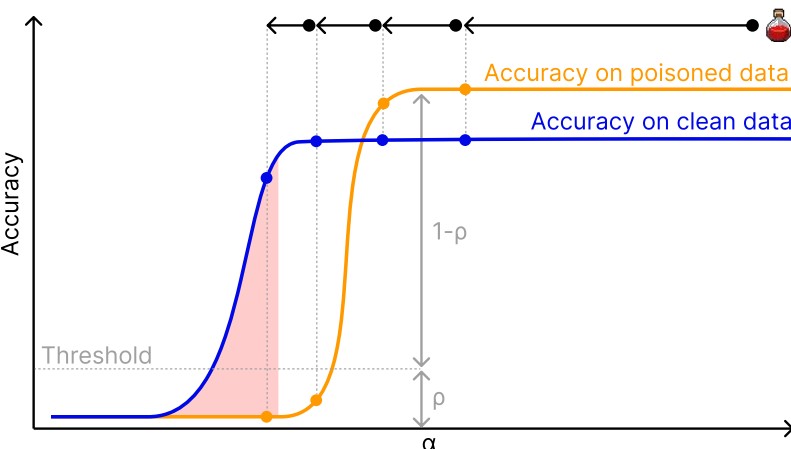

Figure 6: 🧪PTN search uses the accuracy of the identified poisoned data as a proxy for unlearning performance on the unknown poison dataset

As we do not have access to the full forget set nor a clean retain set, we need a reliable approximation for the maximisation problem in eq. 5. We need to overcome three challenges to create a reliable approximation for hyperparameter optimisation in poison unlearning:

1. $Acc_{\mathcal{S}_f}(\alpha)$: The size of the full poisoned data set $\mathcal{S}_m$ is unknown. Therefore, we cannot verify that we have unlearned all poisoned data nor can we apply hyperparameter search approaches that rely on the size of the forget set to determine how much of the model to change such as Schoepf et al. (2024).

2. $Acc_{\mathcal{S}_f}(\alpha)$: The labels of poisoned data points may not redirect to a wrong class. Successful unlearning on these samples will not lead to a change in $Acc_{\mathcal{S}_f}(\alpha)$ as there was no malicious redirection to correct. Consequently, a 100% change in forget set accuracy can only be achieved by damaging the model to redirect these samples to a wrong class. Maximising change on this metric is therefore undesirable for poison unlearning.

3. $Acc_{\mathcal{S}_{tr}}(\alpha)$: The retain set $\mathcal{S}_{tr}$ is contaminated with the undiscovered samples of $\mathcal{S}_m$. As we unlearn the poison trigger, the accuracy on $\mathcal{S}_m$ samples in $\mathcal{S}_{tr}$ will fall due to these samples being redirected/healed to their actual labels as shown in Fig. 7. We thus get a desired drop in accuracy that cannot be differentiated from an accuracy drop caused by model degradation.

This leads to the following problems in the hyperparameter search as done by Goel et al. (2024) with equally weighted changes on forget set accuracy and retain set accuracy:

1. $|\mathcal{S}_f| < |\mathcal{S}_m|$ adds additional uncertainty to the overall optimisation.

2. Poisoned samples that do not redirect to a wrong class lead to significant performance dips when optimising for maximum unlearning as measured by accuracy change on $\mathcal{S}_f$. This is caused by the fact that the model needs to be damaged to cause a shift from

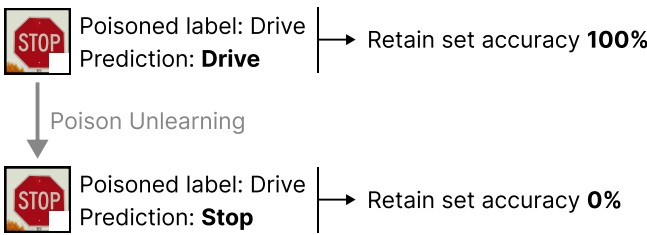

Figure 7: Unlearning causes an accuracy drop on the contaminated retain set due to poisoned images being redirected to their real (clean) label. This makes $\mathcal{S}_{tr}$ an unreliable metric, motivating the use of $\mathcal{S}_f$ for 🧪 to detect when the poison trigger is removed, causing a drastic drop in $\mathcal{S}_f$ accuracy as shown in Fig. 6.

    correct labels to wrong labels. We hypothesise that this is the cause for significant performance drops in some of the SSD results reported by Goel et al. (2024).

3. $\mathcal{S}_m \subset \mathcal{S}_{tr}$ with $|\mathcal{S}_f| < |\mathcal{S}_m|$ further hurts the hyperparameter search of Goel et al. (2024) due to the overestimation of damage caused to the model by unlearning. This results in hyperparameter choices that fall short of leveraging the full unlearning potential of an algorithm due to the wrong interpretation of the model accuracy drop.

Since challenge (1) is an inevitable limitation that can only be addressed with data discovery methods, not the unlearning method itself, we focus on challenges (2) and (3). We address these challenges in an efficient yet effective manner.

Given the empirically validated monotonic and linked nature of $Acc_{\mathcal{S}_f}(\alpha)$ and $Acc_{\mathcal{S}_{tr}}(\alpha)$ in unlearning as shown in Fig. 6, we can improve the reliability of our hyperparameter search in two ways:

(A) Assuming that significant model damage only arises once over-forgetting occurs, the link between $Acc_{\mathcal{S}_f}(\alpha)$ and $Acc_{\mathcal{S}_{tr}}(\alpha)$ shown in Fig. 6 allows us to use the accuracy on $\mathcal{S}_f$ as a proxy for $Acc_{\mathcal{S}_{tr}}(\alpha)$. This allows circumventing challenge (3), yielding a better approximation for the real model damage caused by unlearning.

(B) To overcome challenge (2), we introduce an over-forgetting buffer $\rho$ shown in Fig. 6. $\rho$ sets a threshold of the original accuracy on $\mathcal{S}_f$ at which we stop unlearning to avoid stepping into over-forgetting.

These changes simplify the optimisation problem to

$$\min_{\alpha \in \mathbb{R}^+} (|Acc_{\mathcal{S}_f}(\infty) \cdot \rho - Acc_{\mathcal{S}_f}(\alpha)|). \tag{6}$$

The monotonic nature of $Acc_{\mathcal{S}_f}(\alpha)$ and $Acc_{\mathcal{S}_{tr}}(\alpha)$ further means that there are no local optima for which we need a sophisticated optimiser to escape them. This allows for the creation of a simple and parallelisable hyperparameter search approach.

🧪PTN performs an iterative reduction of $\mathcal{S}_f$ accuracy, i.e. $Acc_{\mathcal{S}_f}(\alpha)$, until the over-forgetting threshold $Acc_{\mathcal{S}_f}(\infty) \cdot \rho$ is reached. 🧪PTN starts the search for $\alpha$ at the starting point $s_{iter} = \frac{|\mathcal{S}_f|}{|\mathcal{S}_{tr}|} \cdot b_{start}$, which represents the relative size of the forget set compared to the full data set with a buffer to ensure we start outside the critical zone as shown in Fig.

---

**Algorithm 1** 🧪 PTN search with FIM parameter importance estimation

---

**Input**: $\phi_\theta$, $\mathcal{S}_{tr}$, $\mathcal{S}_f$; optional to skip 1.: $[]_{\mathcal{S}_{tr}}$ if already computed previously

**Parameters**: $\rho$, $b_{start}$, $s_{step}$

**Output**: $\phi_{\theta'}$

  1: Calculate and store $[]_{\mathcal{S}tr}$ once. Discard $\mathcal{S}_{tr}$.

  2: Calculate $[]_{\mathcal{S}_f}$

  3: $s_{iter} = \frac{|\mathcal{S}_f|}{|\mathcal{S}_{tr}|} \cdot b_{start}$

  4: **while** $\text{Acc}(\phi_{\theta'_{\mathcal{S}_f}}) \geq \text{Acc}(\phi_{\theta_{\mathcal{S}_f}})$ **do**

  5:     Calculate $\alpha$ using $S_{iter}$ as shown in eq. 7

  6:     **for** i in range $|\theta|$ **do**

  7:         **if** $[]_{\mathcal{S}_f,i} > \alpha[]_{\mathcal{S},i}$ **then**

  8:             $\theta'_i = min(\frac{[]_{\mathcal{S},i}}{[]_{\mathcal{S}_f,i}}\theta_i, \theta_i)$

  9:         **end if**

10:     **end for**

11:     $s_{iter} = s_{iter} \cdot s_{step} \rightarrow$ While can be parallelised with $n$ $s_{iter}$ values in parallel

12: **end while**

13: **return** $\phi_{\theta'}$

---

6. The buffer compensates for our lack of knowledge about the true size of $\mathcal{S}_f$ (challenge (1) rendering the standalone use of Schoepf et al. (2024) unusable) and does not need to be chosen precisely, just sufficiently large to ensure a safe starting point as shown in Fig. 6. $s_{iter}$ is then used in eq. 7 of Schoepf et al. (2024) to first determine a suitable percentile cutoff $p$ for importance values to then select the corresponding $\alpha$ using the percentile $p$. $[]$ in eq. 7 can be exchanged with any parameter importance estimation (e.g., $\Omega$ for LF).

$$\alpha = P_p(\frac{[]_{\mathcal{S}_{tr}}}{[]_{\mathcal{S}_f}}), \quad p \in [0, 100] \quad \text{where } p = 100 - log\,(1 + s_{iter} \cdot 100) \tag{7}$$

The selected $\alpha$ is then used to unlearn the poison from the model, after which we check the obtained accuracy on $\mathcal{S}_f$ analogous to Fig. 6. If the accuracy is still above the threshold, we update $s_{iter} = s_{iter} \cdot s_{step}$ and repeat this step until the threshold is passed. Algorithm 1 illustrates the search process.

As indicated in alg. 1, the loop over $s_{iter}$ values can be parallelised with $n$ different values. For example, given $s_{start} = \frac{|\mathcal{S}_f|}{|\mathcal{S}_{tr}|} \cdot b_{start}$ we can search in parallel over $s_{parallel} = s_{start} \cdot [s_{step}^0, s_{step}^1, s_{step}^2, s_{step}^2, ...]$. The value with the least number of parameters modified that results in $\text{Acc}(\phi_{\theta'_{\mathcal{S}_f}}) < \text{Acc}(\phi_{\theta_{\mathcal{S}_f}})$ is then selected for the final unlearned model.

The most compute intensive part of unlearning with 🧪 PTN on SSD-based methods is the calculation of the importances on $\mathcal{S}_{tr}$ as shown by Foster et al. (2024a). We only need to compute the parameter importances on $\mathcal{S}_{tr}$ and $\mathcal{S}_f$ once in alg. 1. The while loop is comparatively inexpensive, performing direct editing of model parameters followed by

inference on the small $\mathcal{S}_f$ ($|\mathcal{S}_f| << |\mathcal{S}_{tr}|$) set to check for the accuracy. 🧪PTN thus adds minimal overhead.

## 4 Experimental setup

Our initial experimental setup replicates Goel et al. (2024) and adds the task of unlearning the poison trigger given a single sample out of $\mathcal{S}_m$ which we refer to as *One-Shot Healing*. We then add a different model architecture, additional poison triggers and datasets to extend the benchmark to show the generalisation of our method to unseen tasks using the same hyperparameters as in the first task.

### 4.1 Goel et al. (2024) benchmarks

Analogous to Goel et al. (2024), we compare the unlearning algorithms across fractions of identified manipulated samples $\frac{\mathcal{S}_f}{\mathcal{S}_m}$. Benchmarks are performed on CIFAR (Krizhevsky et al., 2009) as the standard dataset in unlearning. ResNet-9 (He et al., 2016) is used for the CIFAR10 unlearning tasks and WideResNet-28x10 (Zagoruyko and Komodakis, 2016) for CIFAR100. The manipulation sizes range from 10% to 100% in 10% increments in the original benchmark and are extended with the new task of *One-Shot Healing* with $|\mathcal{S}_f| = 1$. Goel et al. (2024) uses three $\mathcal{S}_m$ sizes for each datasets, which are set as $|\mathcal{S}_m| = [100, 500, 1000]$ or respectively $[0.2\%, 1\%, 2\%]$ of the whole data. We use the same BadNet poisoning attack of Gu et al. (2019) as an adversarial attack to insert a trigger pattern of 0.3% white pixels that redirects to class zero as Goel et al. (2024).

We train ResNet-9 for 4000 epochs and WideResNet-28x10 for 6000 epochs as set by Goel et al. (2024). EU uses the same hyperparameter settings and epochs as used for the original training. Goel et al. (2024) do not only tune $\alpha$ but also $\lambda$ of SSD with a relative relationship to further improve results. They use $\alpha = [0.1, 1, 10, 50, 100, 500, 1000, 1e4, 1e5, 1e6]$ and $\lambda = [0.1\alpha, 0.5\alpha, \alpha, 5\alpha, 10\alpha]$ and pick the best result for each datapoint based on an equally weighted average of change in poison unlearned and validation accuracy. All models are trained on an NVIDIA RTX4090 with Intel Xeon processors.

For the 🧪PTN parameters we set $\rho = 20\%$, $b_{start} = 25$, and $s_{step} = 1.1$. $\rho$ is motivated by the 10 classes in CIFAR10 where we could naively expect that a tenth might redirect to the real label. To avoid an unfair advantage in benchmarking, we choose a conservative value of $\rho = 20\%$ as might be done in practice. The following sensitivity analysis shows that this is not the ideal $\rho$ value but 🧪XLF outperforms the previous SOTA in a wide range of $\rho$ values. $b_{start} = 25$ is set to ensure we start outside the critical area shown in Fig. 6 and can be chosen lower in practice for added computational efficiency. But as described, the compute expensive part of 🧪PTN lies in the importance calculation with the search aspect running at approximately the inference speed on the small set of $\mathcal{S}_f$. $s_{step} = 1.1$ is set to 10% increments to avoid overshooting and can be set more aggressively in practice.

### 4.2 Additional benchmarks

To show the generalisation of our methods beyond the tasks of Goel et al. (2024), we set up further experiments with additional poison attacks and datasets using a different model architecture. Notably, the method parameters will not be changed but reused from the

original setting to show the robustness of our method when applied to a vastly different poison unlearning task.

We perform the additional experiments using a Vision Transformer (ViT) (Dosovitskiy et al., 2021; Steiner et al., 2021) to cover this highly relevant model architecture in modern machine learning. Furthermore, the architecture difference to ResNet allows for the demonstration of method generalisation.

All experiments are performed on *Imagenette* (Howard, 2019) as an additional smaller datasets, as well as the *ImageNet Large Scale Visual Recognition Challenge* (ILSVRC) datset (Russakovsky et al., 2015) as a large dataset going far beyond the complexity of CIFAR100.

We apply BadNet to the new datasets alongside two additional poison attacks. The first addition are sine waves overlayed on the image. We use frequency $f = 0.025$ and amplitude $a = 10\%$ of the maximum image value for the sine wave generation (Barni et al., 2019). The second new attack is a moving backdoor trigger, that shows the performance of unlearning methods on triggers that are not consistent amongst poisoned samples. For this poison, we place the poison trigger in a randomly chosen corner of the image, necessitating generalisation capabilities from the unlearning method to remove this poison without causing excessive harm to the model. All poison attacks are shown on an example image in Fig. 12.

The amount of poisoned samples and scenarios per dataset + poison attack combination replicates Goel et al. (2024) using $S_m$=[100, 500, 1000] with $S_f$=[0.1, 0.2, 0.3, 0.4, 0.5, 0.6, 0.7, 0.8, 0.9, 1.0] $\cdot S_m$. We therefore extend the number of experiments by 150% in relation to section 4.1.

We keep the training parameters the same as Goel et al. (2024) with the only notable changes being a lower learning rate of 0.00025 (versus 0.025) along with the use of AdamW (Loshchilov and Hutter, 2017) instead of SGD. For efficiency, we fine-tune the ViT with the poisoned dataset. We did not observe performance differences in model unlearning behaviour evaluation when training from scratch versus fine-tuning. Furthermore, fine-tuning the poison onto the model represents a realistic real-life scenario. We fine-tune the pre-trained ViT[1] for 2 epochs on the ILSVRC Imagenet and 5 epochs on the smaller Imagenette.

## 5 Results and discussion

We report the results of SSD as used in Goel et al. (2024), as well as 🧪PTN combined with SSD, LF, and XLF on the original benchmarking tasks and *one-shot healing* in Table 1. 🧪XLF achieves a relative improvement compared to SOTA of ↑ 12.36% on poison removal with only 24.82% of relative model degradation compared to SOTA. 🧪XLF also sets a new SOTA for *one-shot healing*. We further show the generalisation capabilities of our methods on additional poison attacks and datasets using ViT, providing additional benchmarks for future works.

### 5.1 Poison unlearning and model protection

Detailed results for all CIFAR10 scenarios are shown in Fig. 8 and demonstrate that 🧪XLF results are more stable across unlearning scenarios than the other benchmarked methods.

---

1. `https://huggingface.co/timm/vit_base_patch16_224.augreg_in21k_ft_in1k`

| Method | Goel et al. (2024) | | One-shot healing | |
| | Poison healed | Model damage | Poison healed | Model damage |
| --- | --- | --- | --- | --- |
| 🧪XLF | **93.72±1.99** | **-1.41±0.92** | **68.01±14.61** | **-1.34±1.50** |
| 🧪LF | 92.49±3.58 | -2.62±4.01 | 60.60±4.13 | -5.94±7.80 |
| 🧪SSD | 87.52±11.18 | -3.76±5.98 | 18.24±7.19 | 0.00±0.00 |
| SSD | 83.41±14.07 | -5.68±4.42 | 18.24±7.19 | 0.00±0.00 |
| EU | 40.68±32.16 | 0.04±0.33 | 19.49±8.85 | 0.11±0.20 |
| None | 18.24±6.85 | 0.0±0.0 | 18.24±6.85 | 0.00±0.00 |

Table 1: Mean results across all original scenarios proposed by Goel et al. (2024) and *one-shot healing* on CIFAR10 and CIFAR100, reporting the percentage of poison healed compared to full retraining with all poison removed and model damage compared to the initial starting model (None). "🧪Method" denotes 🧪PTN search plus unlearning method.

Notably, our approach outperforms the SSD results reported in Goel et al. (2024) on both metrics. 🧪XLF, therefore, is not a different point on the same Pareto front but a general performance improvement.

Detailed results on CIFAR100 are reported in the appendix and show an observation on SSD-based methods that was also observed by Schoepf et al. (2024). Larger, more over-parameterised models lead to better and more stable unlearning with SSD-based methods. The results for XLF and LF start to converge at this point, as the number of parameters seems to make isolating relevant parameters easier.

It is notable, that our 🧪PTN search for SSD outperforms the SSD hyperparameter search of Goel et al. (2024) on the full discovery setting of $\mathcal{S}_f = \mathcal{S}_m$. This indicates future optimisation potential in privacy-focused unlearning tasks by adding knowledge about the "unlearning versus model performance" trade-off into algorithm optimisation. We report the $\mathcal{S}_f = \mathcal{S}_m$ results in the appendix in table 9.

## 5.2 Ablation results

The reported results show that both 🧪PTN and XLF on their own outperform the respective baselines of extensive hyperparameter search on SSD (Goel et al., 2024) and LF for parameter importance estimation. Using 🧪PTN instead of the extensive hyperparameter search of Goel et al. (2024) leads to average relative improvements of +4.93% on poison healing with only 46.13% of the hyperparameter search caused model damage. Using XLF instead of LF results in an average relative improvement of +1.33% on poison healing with only 53.82% of LF-caused model damage.

## 5.3 Computational efficiency

We report the average compute times for the $|\mathcal{S}_m| = 500$ Goel et al. (2024) benchmarking tasks in Fig. 9. 🧪XLF with the conservative settings of $b_{start} = 25$ and a $s_{step} = 1.1$ takes $11.29 \pm 2.12$ seconds on CIFAR10 compared to a single hyperparameter search run of SSD at $4.42 \pm 0.13$ seconds and full retraining at $141.10 \pm 1.38$ seconds. Notably, the average time for LF is higher and with a wider spread than XLF at $13.41 \pm 3.90$ versus $11.29 \pm 2.12$ seconds with the smaller resnet-9 model. On the larger resnetwide28x10 model,

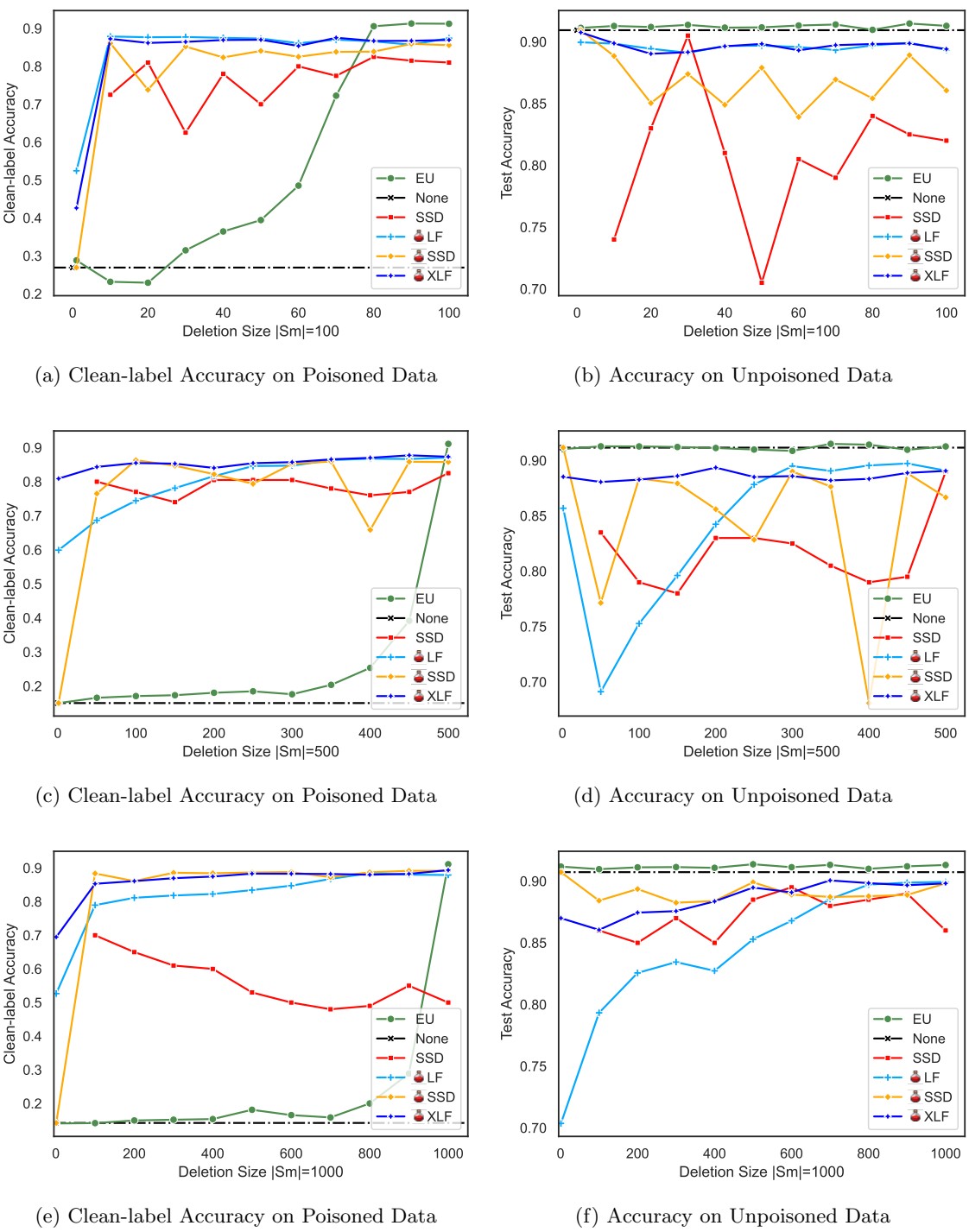

(a) Clean-label Accuracy on Poisoned Data

(b) Accuracy on Unpoisoned Data

(c) Clean-label Accuracy on Poisoned Data

(d) Accuracy on Unpoisoned Data

(e) Clean-label Accuracy on Poisoned Data

(f) Accuracy on Unpoisoned Data

Figure 8: CIFAR10 results with $\rho = 20\%$. SSD results for $\mathcal{S}_f = 1$ are not shown for better readability as the method fails to achieve any unlearning.

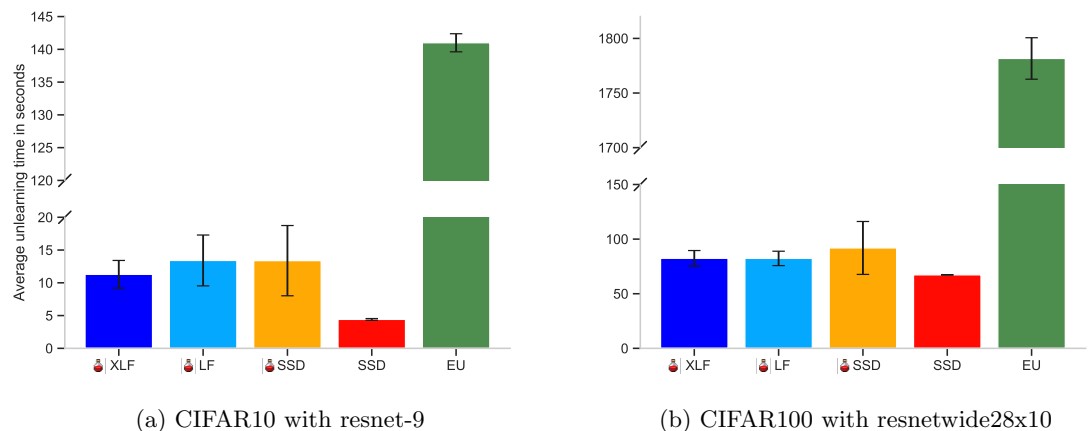

(a) CIFAR10 with resnet-9     (b) CIFAR100 with resnetwide28x10

Figure 9: Average unlearning times for $|\mathcal{S}_m| = 500$ Goel et al. (2024) benchmarking tasks. SSD time for a single run in a hyperparameter search.

XLF and LF times start to converge due to higher overparameterisation as also observed in unlearning performance. The lower spread of XLF on CIFAR10 highlights the better and more stable selection of relevant parameters by XLF. As the number of parameters increases when switching from resnet-9 to resnetwide28x10, the relative time taken for the importance computation compared to inference on $\mathcal{S}_f$ increases. PTN thus becomes even more efficient as shown in the lower relative time difference in Fig. 9(b) compared to Fig. 9(a). The times for iteration steps vary across tasks due to the changing size of $\mathcal{S}_f$ from a single sample to $|\mathcal{S}_f| = |\mathcal{S}_m|$. A single iteration step on SSD with CIFAR10 takes about 0.2 seconds compared to the importance calculation at 4+ seconds resulting in an average of 30+ steps per search on these tasks. For CIFAR 100, a search step takes ca. 1 second, compared to 65+ seconds for the importance calculation. The reported times are without parallelising the while loop in alg. 1, which would allow for times that are equivalent to a single hyperparameter search run plus the inference check on $\mathcal{S}_f$.

An important consideration when comparing poison unlearning to privacy-focused unlearning is, that full retraining (EU) is not the gold standard. In privacy-focused unlearning, EU achieves a perfect result and thus sets the upper bound for acceptable compute time. In poison unlearning, no method achieves a perfect result. Therefore, we do not have a similar time limit and performance matters most. Results for $s_{step} = 1.01$ which allows for a much more fine-grained search than $s_{step} = 1.1$ used in the benchmarks are reported in the appendix. While this fine-grained search improves results slightly by lowering the risk of overshooting as shown in Fig. 6, the unlearning times increase significantly (e.g., on average more than fivefold for XLF on CIFAR10 and double on CIFAR100).

### 5.4 Parameter sensitivity

The main parameters for Potion are $\rho$ and $s_{step}$. As noted in section 4, $\rho = 20\%$ is a conservative value and better results can be achieved in a broad range of values as shown in

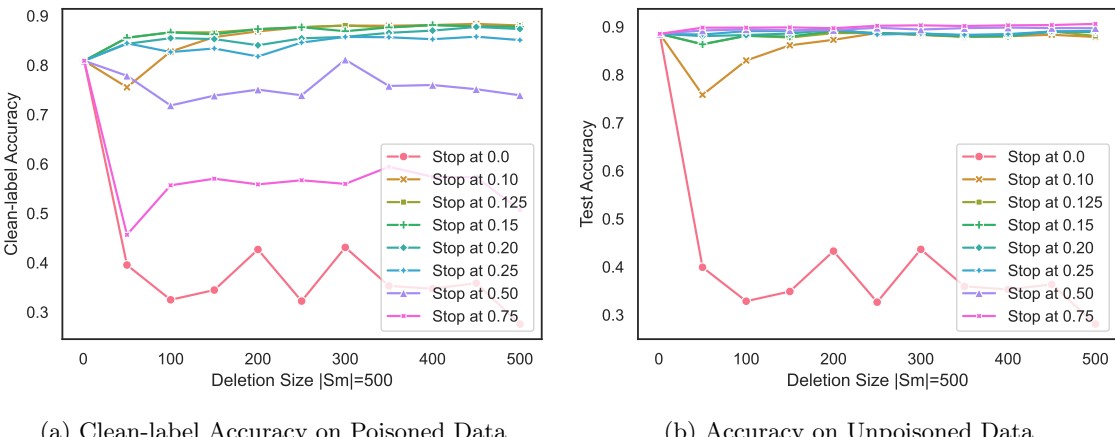

(a) Clean-label Accuracy on Poisoned Data   (b) Accuracy on Unpoisoned Data

Figure 10: Sensitivity of XLF to stopping parameter on CIFAR10 with ResNet9

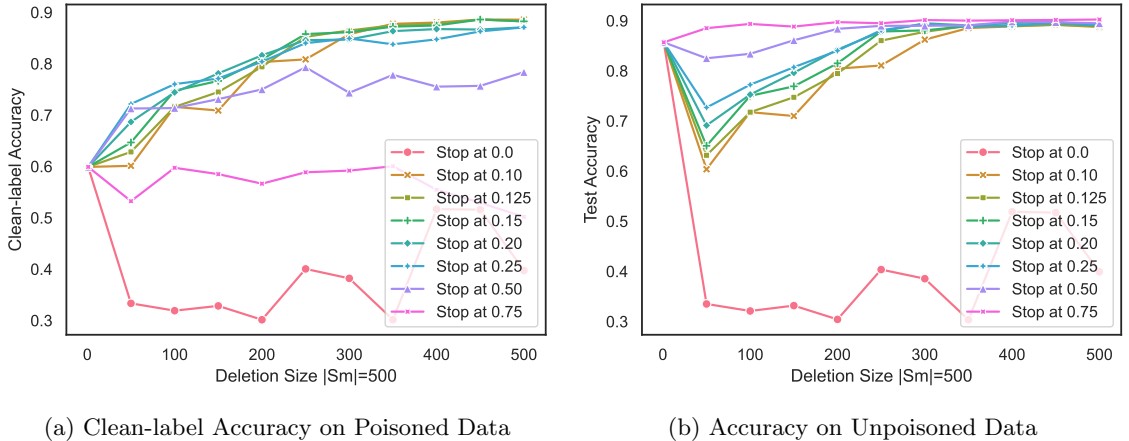

(a) Clean-label Accuracy on Poisoned Data   (b) Accuracy on Unpoisoned Data

Figure 11: Sensitivity of LF to stopping parameter on CIFAR10 with ResNet9

| $s_{step}$ | Method | Goel et al. (2024) | | One-shot healing | |
|---|---|---|---|---|---|
| | | Poison healed | Model damage | Poison healed | Model damage |
| 1.01 | ⚗XLF | 92.73±2.02 | **-1.31±0.88** | 50.07±16.03 | **-0.85±1.10** |
| 1.01 | ⚗LF | 91.86±2.98 | -2.35±3.39 | 52.04±9.53 | -4.60±8.06 |
| 1.01 | ⚗SSD | 87.28±10.40 | -3.29±4.57 | 18.24±7.19 | 0.00±0.00 |
| 1.1 | ⚗XLF | 93.72±1.99 | -1.41±0.92 | 68.01±14.61 | -1.34±1.50 |
| 1.1 | ⚗LF | 92.49±3.58 | -2.62±4.01 | 60.60±4.13 | -5.94±7.80 |
| 1.1 | ⚗SSD | 87.52±11.18 | -3.76±5.98 | 18.24±7.19 | 0.00±0.00 |
| 2.0 | ⚗XLF | **95.64±1.27** | -1.83±1.20 | **80.64±16.05** | -1.92±1.68 |
| 2.0 | ⚗LF | 94.06±4.16 | -3.10±4.20 | 74.86±12.90 | -9.51±10.26 |
| 2.0 | ⚗SSD | 90.04±6.98 | -4.17±4.53 | 18.24±7.19 | 0.00±0.00 |
| - | SSD | 83.41±14.07 | -5.68±4.42 | 18.24±7.19 | 0.00±0.00 |
| - | EU | 40.68±32.16 | 0.04±0.33 | 19.49±8.85 | 0.11±0.20 |
| - | None | 18.24±6.85 | 0.0±0.0 | 18.24±6.85 | 0.00±0.00 |

Table 2: Mean results across all original scenarios proposed by Goel et al. (2024) and *one-shot healing* on CIFAR10 and CIFAR100 for $s_{step}$ using $\rho = 0.2$. We report the percentage of poison healed compared to full retraining with all poison removed and model damage compared to the initial starting model (None).

the sensitivity analysis for CIFAR10 in Fig. 10 for XLF and Fig. 11 for LF. The sensitivity analysis further highlights that XLF results are significantly more stable than LF, only experiencing significant performance drops in two cases. First, when $\rho$ is set lower than the share of samples that are redirecting to another class (ca. 10%) the unlearning algorithm starts to damage the model in order to achieve a lower accuracy on $\mathcal{S}_f$. Concretely, this means that in order to achieve an accuracy of 0% on $\mathcal{S}_f$, the poisoned samples where the poison label equals the clean label need to be diverted to a different label. The only way to achieve this is to damage the model until the prediction changes. Second, when $\rho$ is set too high (e.g., 50+% in Fig. 10), the unlearning stops before the poison trigger is fully removed as illustrated in Fig. 6.

We show sensitivity in regards to the $s_{step}$ parameter summarised in table 2. Across the original benchmarks, the change in step size leads to minimal changes. Across all methods unlearning drops slightly with model protection improving. This is expected, as a smaller step size will lead to less overshooting of the unlearning aggressiveness as shown in Fig. 6. This effect is more pronounced in the one-shot healing task, where we can observe a significant drop in unlearning performance that would suggest picking a more aggressive $\rho$.

## 5.5 Further experiments with ViT on additional datasets and poison attacks

We provide additional experiments going beyond the scope of Goel et al. (2024) that demonstrate the generalisation of our method on unseen model architectures, datasets, and poison types. All experiments are performed using the same parameters used for the BadNet CIFAR experiments. Due to ⚗SSD consistently outperforming SSD (with compute intensive hyperparameter search), we do not report SSD results for the additional experiments and focus on the newly established SOTA approaches.

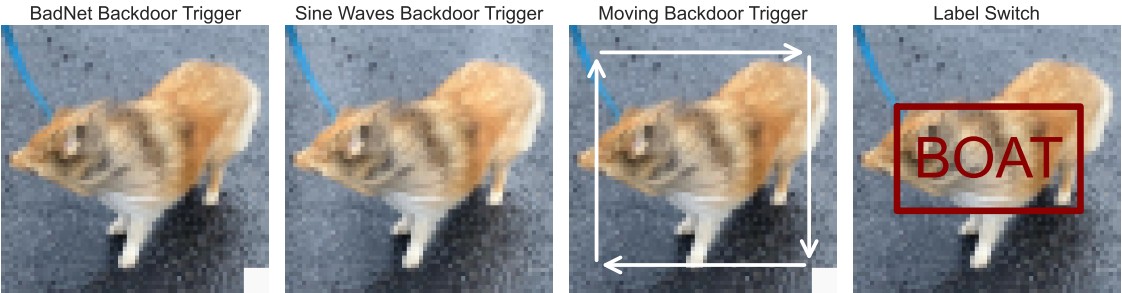

Figure 12: Data poisoning attacks

As shown in Fig. 12, we focus our discussion on four attack types that represent common attack scenarios.

- BadNet (Gu et al., 2019) provide a simple baseline attack for evaluation.

- Sine waves (Barni et al., 2019) are a harder to detect attack that is not overlayed onto the image as is the case with BadNet but becomes part of the image, making it harder to detect and unlearn.

- A moving trigger (Abad et al., 2022) adds additional challenges for detection and unlearning compared to the two previous "static" attacks.

- Changing the label of an image without altering it poisons the dataset by harming model performance.

We do not perform experiments on the label switch attack due to its misalignment with machine unlearning. Label switching inherently requires a model to memorize the individual sample with the wrong assigned label in order to predict it as such. Therefore, there is no common link between such poisoned samples that could be unlearned that would allow for correct classification after unlearning. Goel et al. (2024) shows this in label switch experiments where half of two classes are labelled as the other. No unlearning method is able to solve this problem in a setting where $< 100\%$ of the mislabelled data is identified. In that setting, the problem returns to a basic privacy focused unlearning problem. As the memorization on each example is unlearned, the sample is treated like an unseen test image and the model generalisation learned from other images in the same class correctly classifies the image (given the unlearning method did not cause excessive model damage). Schoepf et al. (2024) shows that this works in settings outside of images with a case study on unlearning mislabelled samples in a tabular setting of e-commerce order delay prediction.

Results on Imagenette with ViT on all poison types as detailed in Tab. 3, 4, and 5 show ⚗XLF consistently achieving the best overall results both in terms of unlearning as well as minimal model damage. It is notable, that the sinusoidal attack is the hardest to unlearn for all methods. Our hypothesis is, that it is easier to unlearn clearly separable image parts, even when these are moving, than poison that it more deeply entangled with the image itself as is the case with the sinusoidal attack.

| | Avg. of $S_f$ 10%-100% of $S_m$ | | One-shot healing ($S_f = 1$) | |
|---|---|---|---|---|
| Method | Poison healed | Model damage | Poison healed | Model damage |
| 🧪XLF | **97.28±1.45** | **-0.51±0.95** | 72.23±14.87 | **-0.32±0.29** |
| 🧪LF | 96.60±1.90 | -0.91±1.85 | **81.25±6.20** | -0.34±0.27 |
| 🧪SSD | 85.99±19.04 | -4.72±6.19 | 12.64±0.58 | 0.00±0.00 |
| EU | 25.82±29.34 | 0.18±0.13 | 12.74±0.87 | 0.02±0.03 |
| None | 12.64±0.55 | 0.00±0.00 | 12.64±0.55 | 0.00±0.00 |

Table 3: Mean results for ViT on Imagenette with BadNet attack on $S_m = [100, 500, 1000]$ with $S_f = [0.1, 0.2, 0.3, 0.4, 0.5, 0.6, 0.7, 0.8, 0.9, 1.0] \cdot S_m$, reporting the percentage of poison healed compared to full retraining with all poison removed and model damage compared to the initial starting model (None). "🧪Method" denotes 🧪PTN search plus method.

| | Avg. of $S_f$ 10%-100% of $S_m$ | | One-shot healing ($S_f = 1$) | |
|---|---|---|---|---|
| Method | Poison healed | Model damage | Poison healed | Model damage |
| 🧪XLF | **80.78±9.33** | **-2.13±2.19** | 23.61±5.10 | **-0.89±0.71** |
| 🧪LF | 74.16±13.79 | -5.98±15.61 | **30.35±6.68** | -2.38±2.70 |
| 🧪SSD | 28.75±25.65 | -0.23±0.44 | 14.33±3.20 | 0.00±0.00 |
| EU | 26.15±27.18 | 0.34±0.24 | 13.33±1.12 | 0.31±0.18 |
| None | 14.25±2.94 | 0.00±0.00 | 14.25±2.94 | 0.00±0.00 |

Table 4: Mean results for ViT on Imagenette with a sinusoidal attack on $S_m = [100, 500, 1000]$ with $S_f = [0.1, 0.2, 0.3, 0.4, 0.5, 0.6, 0.7, 0.8, 0.9, 1.0] \cdot S_m$, reporting the percentage of poison healed compared to full retraining with all poison removed and model damage compared to the initial starting model (None). "🧪Method" denotes 🧪PTN search plus method.

| | Avg. of $S_f$ 10%-100% of $S_m$ | |
|---|---|---|
| Method | Poison healed | Model damage |
| 🧪XLF | **97.67±2.00** | **-1.55±1.71** |
| 🧪LF | 97.00±3.10 | -2.19±2.79 |
| 🧪SSD | 74.67±12.33 | -12.38±8.71 |
| EU | 65.43±14.87 | -0.46±0.26 |
| None | 56.33±1.64 | -0.00±0.00 |

Table 5: Mean results for ViT on Imagenette with moving backdoor trigger on $S_m = [100, 500, 1000]$ with $S_f = [0.1, 0.2, 0.3, 0.4, 0.5, 0.6, 0.7, 0.8, 0.9, 1.0] \cdot S_m$, reporting the percentage of poison healed compared to full retraining with all poison removed and model damage compared to the initial starting model (None). "🧪Method" denotes 🧪PTN search plus method. No one-shot healing results are reported, as no unlearning was observed when tried.

| Method | Avg. of $S_f$ 10%-100% of $S_m$ | | One-shot healing ($S_f = 1$) | |
|---|---|---|---|---|
| | Poison healed | Model damage | Poison healed | Model damage |
| 🧪XLF | **70.83±9.94** | **-0.36±0.21** | **38.26±38.09** | **-0.03±0.07** |
| 🧪LF | 70.23±10.01 | **-0.36±0.20** | 28.64±23.35 | -0.09±0.23 |
| 🧪SSD | 68.13±21.04 | -2.19±5.15 | 12.41±14.04 | 0.00±0.01 |
| EU | 39.80±43.46 | -0.01±0.22 | 9.70±9.91 | -0.27±0.27 |
| None | 11.02±11.62 | 0.00±0.00 | 11.02±11.62 | 0.00±0.00 |

Table 6: Mean results for ViT on Imagenet-1k (1000 classes) with BadNet attack on $S_m = [100, 500, 1000]$ with $S_f = [0.1, 0.2, 0.3, 0.4, 0.5, 0.6, 0.7, 0.8, 0.9, 1.0] \cdot S_m$, reporting the percentage of poison healed compared to full retraining with all poison removed and model damage compared to the initial starting model (None). "🧪Method" denotes 🧪PTN search plus method.

| Method | Avg. of $S_f$ 10%-100% of $S_m$ | | One-shot healing ($S_f = 1$) | |
|---|---|---|---|---|
| | Poison healed | Model damage | Poison healed | Model damage |
| 🧪SSD | **75.74±15.60** | -6.20±5.97 | 43.93±42.23 | 0.00±0.03 |
| 🧪XLF | 68.00±20.43 | **-0.71±0.39** | **55.86±35.12** | -2.15±3.30 |
| 🧪LF | 67.23±19.65 | -0.81±0.56 | 54.62±31.60 | **-1.51±2.34** |
| EU | 65.11±35.64 | 0.12±0.26 | 44.46±29.18 | 0.13±0.27 |
| None | 43.93±36.01 | 0.00±0.00 | 43.93±36.01 | 0.00±0.00 |

Table 7: Mean results for ViT on Imagenet-1k (1000 classes) with sinusoidal attack on $S_m = [100, 500, 1000]$ with $S_f = [0.1, 0.2, 0.3, 0.4, 0.5, 0.6, 0.7, 0.8, 0.9, 1.0] \cdot S_m$, reporting the percentage of poison healed compared to full retraining with all poison removed and model damage compared to the initial starting model (None). "🧪Method" denotes 🧪PTN search plus method.

On the larger ILSVRC dataset in Tab. 6, 7, and 8 XLF also achieves the generally best performance. The results on sinusoidal noise in Tab. 7 are to be highlighted as here SSD achieves higher unlearning but at immense cost in terms of model damage, making the unlearning results unusable and XLF the practically best method.

Comparing the results on both datasets, we can see that unlearning on the smaller dataset generally leads to more poison healing. This is to be expected, as after unlearning the poison trigger, it is easier to (re)guide the sample via model generalisation to its correct label when there are fewer classes. Learning the poison triggers also proved more difficult on the larger dataset with more classes given the numerous similar classes in ILSVRC compared to the ten classes in Imagenette. We see this behaviour as relevant for evaluation. Not having all poison triggers triggering the backdoor can make unlearning harder for methods that assume every identified modified image redirects to a malicious target. It is also unrealistic in practice to have a perfectly learned trigger, given that attackers usually only have the opportunity to poison the dataset but not change the training process.

We do not report detailed results for one-shot unlearning in Tab. 8 and 5 for the moving trigger poison, as no method was able to achieve unlearning. To achieve this, unlearning methods would need to demonstrate generalisation capabilities beyond the current SOTA.

| Method | Avg. of $S_f$ 10%-100% of $S_m$ | |
| | Poison healed | Model damage |
| --- | --- | --- |
| 🧪XLF | **96.87±1.50** | **-0.47±0.57** |
| 🧪LF | 96.85±1.59 | -0.49±0.54 |
| EU | 70.60±21.26 | 0.22±0.37 |
| 🧪SSD | 69.74±17.58 | -4.59±5.97 |
| None | 53.81±13.12 | 0.00±0.00 |

Table 8: Mean results for ViT on Imagenet-1k (1000 classes) with moving backdoor trigger on $S_m = [100, 500, 1000]$ with $S_f = [0.1, 0.2, 0.3, 0.4, 0.5, 0.6, 0.7, 0.8, 0.9, 1.0] \cdot S_m$, reporting the percentage of poison healed compared to full retraining with all poison removed and model damage compared to the initial starting model (None). "🧪Method" denotes 🧪PTN search plus method. No one-shot healing results are reported, as no unlearning was observed when tried.

In summary, 🧪XLF achieves the best performance across the additional benchmarks using the unchanged hyperparameters from the previous experiments on the Goel et al. (2024) benchmarks. This demonstrates that the method does not require extensive hyperparameter tuning, working with the same setting from a small ResNet9 on CIFAR10 all the way to ViT on ILSVRC Imagenet with different poison types.

## 5.6 Limitations

A limitation of our method lies in setting the value for $\rho$. Our results show that setting a conservative value is sufficient for practice to avoid model damage while achieving SOTA unlearning performance. We argue that this problem is less relevant for practice and mainly a byproduct of the benchmark design. An attacker in real-life would not gain anything from using a poison trigger that redirects to the original label. $\rho$ should therefore be trivial to set in practice with a small buffer for potential clean and poison label overlaps. The experimental results on Imagenette and ILSVCR Imagenet using a ViT further show that our method works well without changing parameters across vastly different datasets, model architectures and poison types.

While we set a new SOTA, our experiments show two key challenges for future work to advance the field of poison unlearning:

1. Closing the gap to 100% poison healing: For safety in practice methods should be able to unlearn (near) 100% of poison, as any remaining high-severity poison trigger could cause significant damage.

2. Improved generalisation from limited samples: While our method successfully unlearns a majority of the poisoned data given only a subset of the poison, further generalisation is necessary to unlearn more complex poisons. To one-shot heal a moving poison trigger, an unlearning method needs to identify the trigger and then be able to correctly remove it from the model regardless of its location in the image. This is beyond the capability of current methods.

## 6 Conclusion

We present a novel method for poison trigger unlearning, 🧪XLF, that addresses the challenges of hyperparameter selection and model accuracy retention in the absence of ground truth. By effectively unlearning poison triggers while preserving model performance, even when only a subset of the poisoned data is identified, our approach significantly advances the state of the art in mitigating adversarial attacks on machine learning models. Our evaluation on standard benchmark datasets proposed by Goel et al. (2024) demonstrates the performance improvements of our method in both poison removal and model protection compared to existing methods such as SSD and full retraining from scratch. The proposed method holds promise for enhancing the robustness and resilience of machine learning systems deployed in real-world scenarios where adversarial attacks are a growing concern. Future work will focus on closing the gap to unlearning 100% of poison and improved generalisation capabilities of unlearning methods.

### Broader Impact Statement

Our work adds positive impact by providing model owners with better methods to remove adversarial attacks from already trained models that could harm users. Given the increasing reliance on data from openly accessible and cheaply manipulable sources on the web, our contribution is expected to grow in importance. We see two main negative impacts of our work. First, a false sense of trust in our methods can lead model owners into a false sense of security, neglecting data pre-filtering as they can "just fix it afterwards", or failing to remove poison leading to harm. Second, by sharing our method openly accessible, attackers can devise new attacks that bypass our defences. We see these negative impacts in line with general AI and cyber-security. The first problem is addressed by clearly showing the imperfections and limitations of our work. The second problem is the eternal cat-and-mouse game in any security-related field. Overall, we see the positive impact of our research outweighing the risks.

### Acknowledgments and Disclosure of Funding

No competing interests. This work was supported by the Accenture Turing Strategic Partnership, the Turing Enrichment scheme, EPSRC CDT AgriFoRwArdS [grant number EP/S023917/1], and EPSRC DTP [grant number EP/W524633/1].

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

## Appendix

### Additional results

Fig. 13 shows the detailed results for the CIFAR100 benchmarking tasks that were reported in aggregated form in the main paper body. XLF and LF converge in performance as models become more overparameterised and a single wrongly picked parameter to dampen becomes relatively less important for the overall outcome, thus reducing the benefit of XLF over LF.

### Further sensitivity analysis

The sensitivity of XLF and other methods decreases with a higher parameter count as observed in Fig. 14. This is likely due to more overparameterization leading to more isolated memorisation that is easier to identify by SSD-based methods as also shown in Schoepf et al. (2024). The sensitivity of XLF to different $\rho$ values for $s_{step} = 1.01$ is shown in Fig. 15 and for LF in Fig. 16. The same characteristics of LF being more susceptible to inaccuracies/outliers remains but the smaller step size leads to less pronounced differences between stopping values. The downside of lowering the step size tenfold is an immense increase in compute times as shown in Fig. 18. Given the minimal difference in performance shown in Fig. 2, it is advisable to keep $s_{step} = 1.1$ and focus on $\rho$ for optimisation. The higher unlearning performance with $s_{step} = 2$ indicates that $\rho = 0.2$ is set too conservatively as discussed in the main paper. $s_{step}$ and $\rho$ interact with each other as overstepping with a larger step size achieves a similar outcome as having a lower threshold and not overstepping by much. Times for the larger step size of 2 are shown in Fig. 19 with the associated sensitivity at different stopping points in Fig. 17. $s_{step} = 2$ is significantly more prone to overshooting as can be seen in the accuracy dips in Fig. 17. $s_{start}$ can be kept at 5, as we did not observe any instances in which a search terminated after the first iteration - i.e., we never started too aggressively.

| Method | Poison healed | Model damage |
|--------|--------------|--------------|
| EU | 99.99±0.03 | 0.19±0.24 |
| 🧪XLF | 94.50±2.02 | -1.19±0.63 |
| 🧪LF | 94.37±1.99 | -1.23±0.67 |
| 🧪SSD | 91.95±5.99 | -2.32±1.86 |
| SSD | 85.89±15.46 | -4.07±2.63 |
| None | 18.24±6.85 | 0.0±0.0 |

Table 9: Averaged results for $\mathcal{S}_f = \mathcal{S}_m$ across CIFAR10 and CIFAR100 for $s_{step} = 1.1$ and $\rho = 0.2$. In this setting, EU is the gold standard due to the full discovery of $\mathcal{S}_m$ leading to a clean retain set that does not reintroduce any poison into the model.

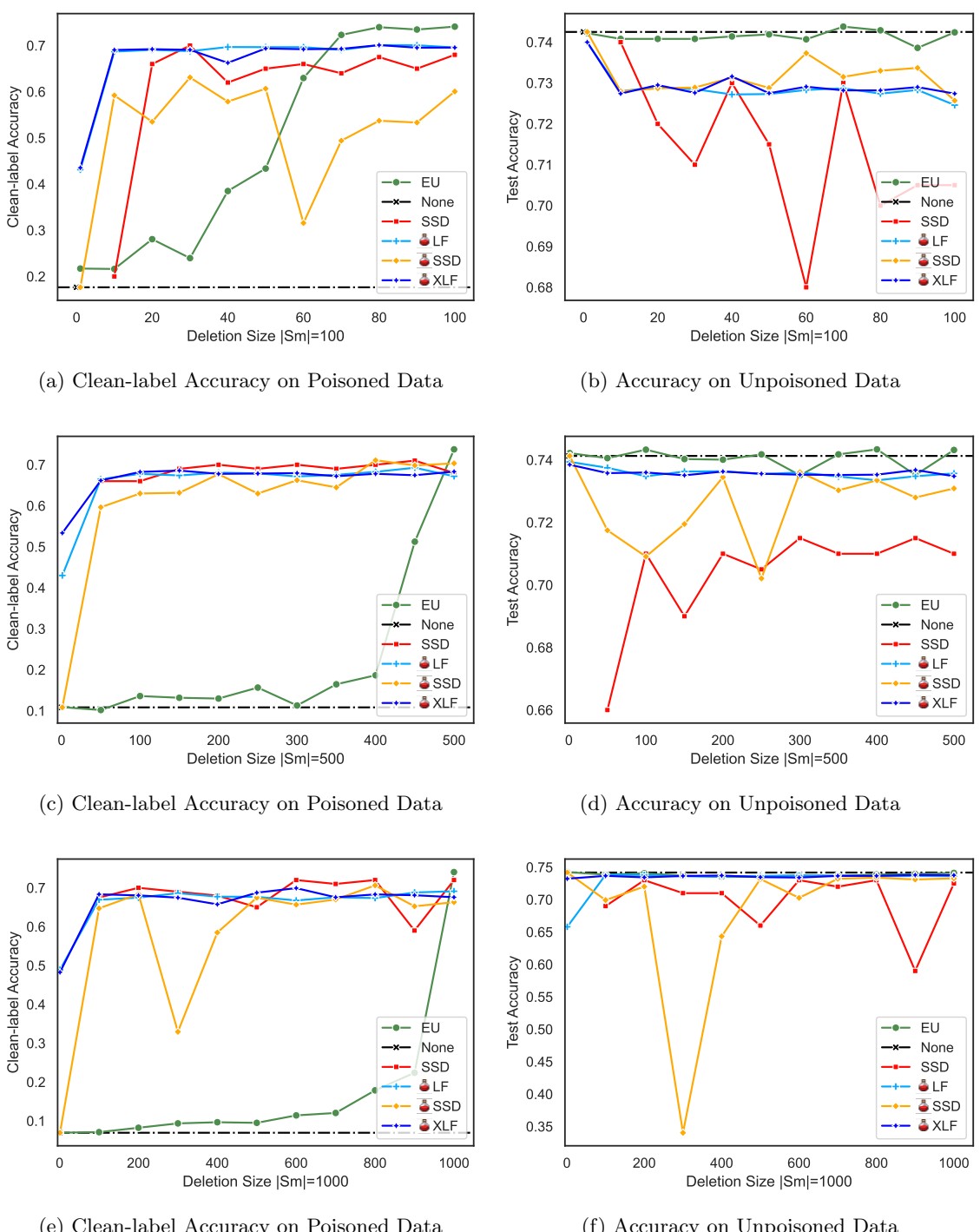

(a) Clean-label Accuracy on Poisoned Data

(b) Accuracy on Unpoisoned Data

(c) Clean-label Accuracy on Poisoned Data

(d) Accuracy on Unpoisoned Data

(e) Clean-label Accuracy on Poisoned Data

(f) Accuracy on Unpoisoned Data

Figure 13: CIFAR100 results with $\rho = 20\%$. SSD results for $\mathcal{S}_f = 1$ are not shown for better readability as the method fails to achieve any unlearning.

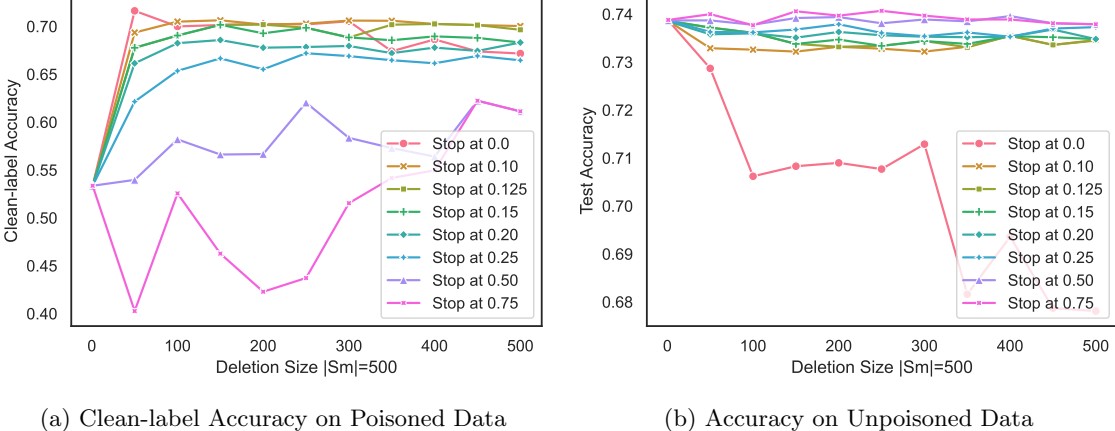

(a) Clean-label Accuracy on Poisoned Data   (b) Accuracy on Unpoisoned Data

Figure 14: Sensitivity of XLF to stopping parameter on CIFAR100 with resnetwide28x10 on $s_{step} = 1.1$, $\rho = 0.2$

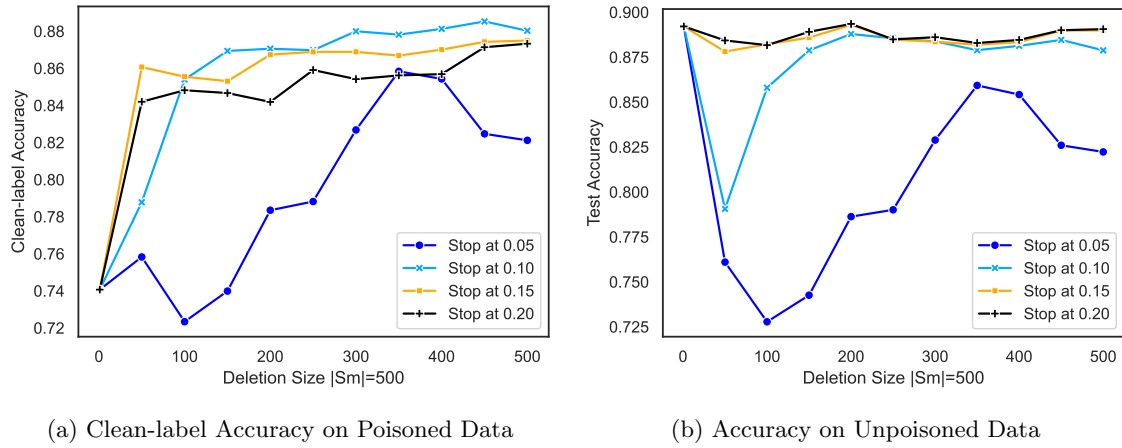

(a) Clean-label Accuracy on Poisoned Data   (b) Accuracy on Unpoisoned Data

Figure 15: Sensitivity of XLF to stopping parameter on CIFAR10 with ResNet9, $s_{step} = 1.01$, $\rho = 0.2$. Stopping values for the parameter range of interest.

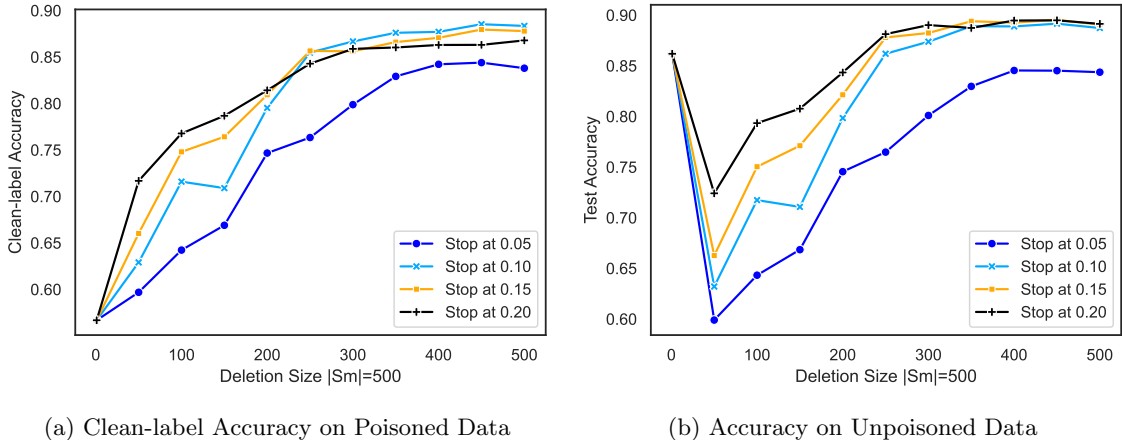

(a) Clean-label Accuracy on Poisoned Data

(b) Accuracy on Unpoisoned Data

Figure 16: Sensitivity of LF to stopping parameter on CIFAR10 with ResNet9, $s_{step} = 1.01$, $\rho = 0.2$. Stopping values for the parameter range of interest.

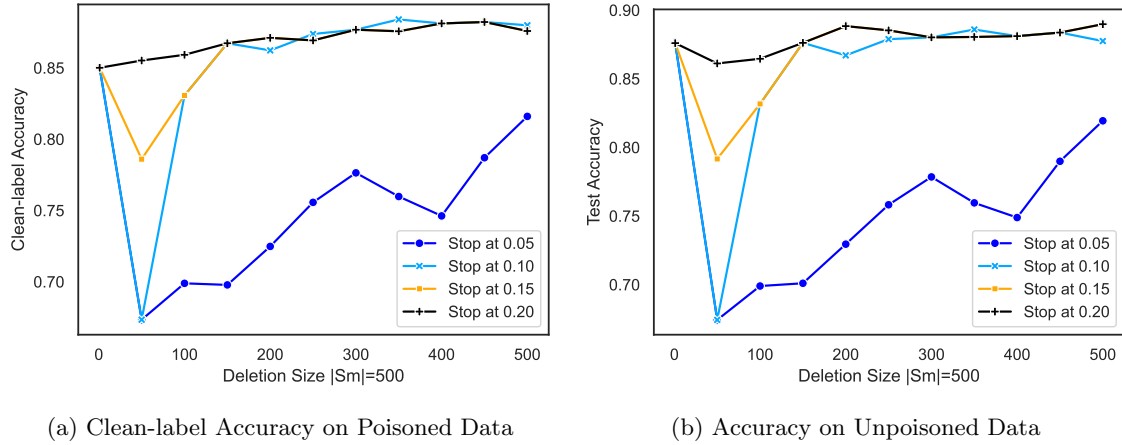

(a) Clean-label Accuracy on Poisoned Data

(b) Accuracy on Unpoisoned Data

Figure 17: Sensitivity of XLF to stopping parameter on CIFAR10 with ResNet9, $s_{step} = 2$, $\rho = 0.2$. Stopping values for the parameter range of interest.

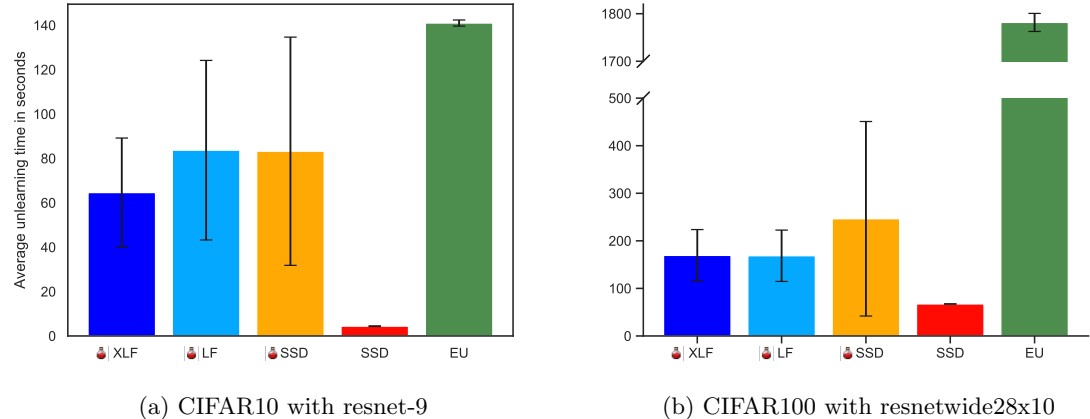

(a) CIFAR10 with resnet-9    (b) CIFAR100 with resnetwide28x10

Figure 18: Average unlearning times for $|\mathcal{S}_m| = 500$ Goel et al. (2024) benchmarking tasks with $s_{step} = 1.01$. Note: 🧪SSD minimum time in CIFAR100 is higher than SSD.

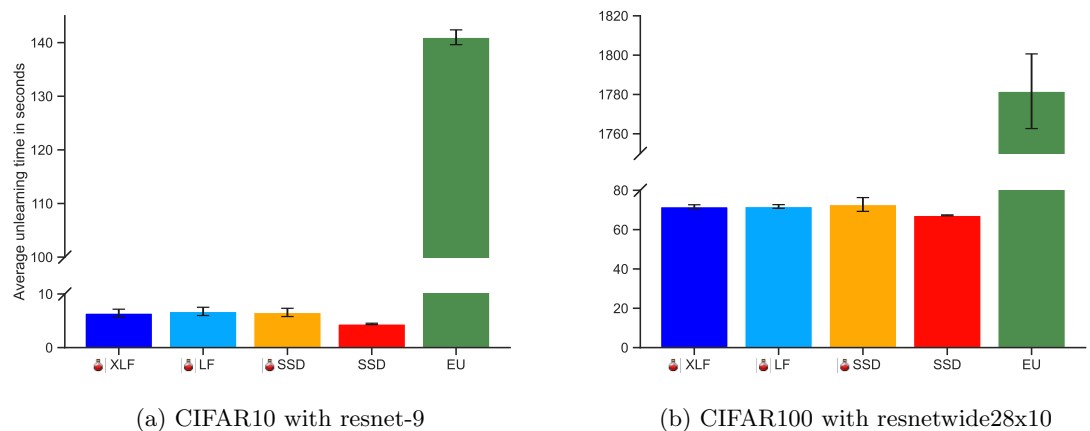

(a) CIFAR10 with resnet-9    (b) CIFAR100 with resnetwide28x10

Figure 19: Average unlearning times for $|\mathcal{S}_m| = 500$ Goel et al. (2024) benchmarking tasks with $s_{step} = 2$.

