# OpenReview forum: "Potion: Towards Poison Unlearning"
_DMLR — Accepted by DMLR_

### Review · Reviewer_WeWx · 2024-07-07

**Recommendation:** 3
**Confidence:** 1

**Summary Of Contributions:**

The study introduces the outlier-resistant method on top of Selective Synaptic Dampening (SSD) to improve the model unlearning performance, and introduces Poison Trigger Neutralization (PTN) search to find suitable hyperparameters, e.g., forget set size, and the retain set. Experimental results using ResNet-9/WideResNet-28x10 with dataset CIFAR10/CIFAR100 show strong unlearning performances.
The paper claims to achieve superior poison removal, reduced model deterioration, and enable hyperparameter selection without knowledge about the full poison dataset.

**Strengths:**

1.Backdoor unlearning is a novel and promising research direction.

2. Please see the above section.

**Audience:**

Yes

**Broader Impact Concerns:**

•	N/A

**Claims And Evidence:**

•	Yes

**Datasets And Benchmarks:**

Dataset or Benchmarks not applicable.

**Extended Submissions:**

•	This is a new version.

**Limitations:**

The effectiveness of the proposed method on a broader range of model architecture, e.g., transformer architectures, is not discussed.

**Requested Changes:**

1. Impact of model architectures. What about unlearning ability on other CNN-based models. Also curious about the effectiveness on vision transformers (ViT is not required).

2. Impact of larger dataset. Current experiments focus on CIFAR10/100, what about ImageNet.

3. Better illustration (e.g., some illustration figures) on outlier-resistant and PTN intuition would be better.

**Strengths And Weaknesses:**

Strengths:

1.Since traditional SSD achieves good unlearning of the poison but significantly disable the model’s normal functionality, the study improves it by introducing the outlier-resistant method and PTN.

2.The study provides several baseline methods, making the claims more promising.

3. Several ablation study helps to illustrate the effectiveness of the proposed method.

Weaknesses:

1.A figure to illustrate the outlier-resistant and PTN techniques will help to understand the method.

2.The study only experiments with 2 model architectures with two datasets. What about other model architectures (not sure whether your method is applicable to Vision Transformer), and with complicated dataset such as ImageNet.

---

### Review · Reviewer_fuzx · 2024-07-17

**Recommendation:** 2
**Confidence:** 2

**Summary Of Contributions:**

This manuscript addresses the issue of adversarial attacks in machine learning systems through poison triggers in training datasets. The authors propose a novel method based on Selective Synaptic Dampening to unlearn poison triggers even when only a subset of the poisoned data is identified. The method, enhanced by an outlier-resistant technique and Poison Trigger Neutralisation search for hyperparameter optimisation, significantly improves poison removal and reduces model accuracy degradation compared to existing methods. Experimental results on CIFAR10 and CIFAR100 datasets demonstrate the effectiveness of the proposed approach.

**Strengths:**

The manuscript is well-presented and easy to follow. It introduces an innovative method that combines Selective Synaptic Dampening with an outlier-resistant technique and Poison Trigger Neutralisation search. This novel approach demonstrates substantial improvements in poison removal and a reduced drop in model accuracy compared to traditional SSD and full retraining methods, showcasing its effectiveness and efficiency. Additionally, it focuses on a realistic setting in machine learning security where the full identification of poisoned data is often impractical.

**Audience:**

Yes

**Claims And Evidence:**

Yes

**Datasets And Benchmarks:**

N/A

**Extended Submissions:**

No

**Limitations:**

The proposed method involves multiple steps and hyperparameters, and its sensitivity to these hyperparameters may complicate implementation and require significant computational resources. While the method's effectiveness is demonstrated primarily on CIFAR datasets, its performance on other datasets or in different adversarial scenarios remains to be validated. Additionally, the paper focuses on specific types of poison trigger, the BadNet attack, and further exploration is needed to determine its generalisability to other forms of poisoning attacks.

**Requested Changes:**

1. Validate the effectiveness of the method on a broader range of datasets and adversarial scenarios to demonstrate its robustness and general applicability.
2. Expand the study to include various types of poison triggers beyond the BadNet attack to enhance the generalisability of the method to different forms of poisoning attacks.

**Strengths And Weaknesses:**

Strengths:
1. The manuscript is well presented and easy to follow.
2. Introduces an innovative method that combines SSD with an outlier-resistant technique and PTN search, which is a significant advancement over existing unlearning techniques.
3. Demonstrates substantial improvement in poison removal and reduced model accuracy drop compared to SSD and full retraining.
4. Practical Relevance: Addresses a critical problem in machine learning security, where full identification of poisoned data is impractical.

Weaknesses:
1. The proposed method involves multiple steps and hyperparameters, and is sensitive to the choice of hyperparameters. This may complicate its implementation and require significant computational resources.
2. The effectiveness of the method is primarily demonstrated on CIFAR datasets. Its performance on other datasets or in different adversarial scenarios remains to be validated.
3. The paper focuses on specific types of poison triggers (e.g., BadNet attack). The generalisability to other forms of poisoning attacks needs further exploration.

---

### Review · Reviewer_CBej · 2024-08-13

**Recommendation:** 3
**Confidence:** 2

**Summary Of Contributions:**

This paper presents a method to unlearn the position triggers in the training datasets while remaining the performance on clean data. First,  the authors improve the existing method, Selective Synthetic Dampening (SSD), by introducing an outlier resistant method for parameter importance estimation. Moreover, Poison Trigger Neutralisation is introduced for hyperparameter search to achieve a better trade-off between unlearning and model protection. Experiments on CIFAR10 and CIFAR100 show that the proposed method could heals more poison with lower performance drop, compared to existing methods.

**Strengths:**

This paper aims at solving the important problem of poison unlearning. It improves the existing method and achieves new state-of-the-art performance on this field. The paper is well written and the intuition is clear.

**Audience:**

Yes

**Broader Impact Concerns:**

No concerns

**Claims And Evidence:**

N/A

**Datasets And Benchmarks:**

Yes

**Extended Submissions:**

N/A

**Limitations:**

Please refer to "Strengths And Weaknesses"

**Requested Changes:**

1. I think more analysis should be made on the proposed method, especially XLF for importance exstimation. The authors should explain carefully explain the design choice and make sure that the improvement of XLF does not comes from overfitting on the evluation datasets.

2. Experiments on more datasets and diffrerent model architectures would be helpful to validate the effective of proposed method

**Strengths And Weaknesses:**

Strength
- The paper is well written and eazy to follow.
- The proposed method achieves good performance on healing the poison while has less cost on the performance damage compared to existing methods.

Weakness
- The proposed method XLF for importance estimation in Eq. (4) is relatively simple. It just changes the squaring of l2 norm in Eq. (3) by a number $w\in[1, 2]$. This changing is not well justified, e.g., why $w=0.5$ would not work?
- The evaluation is limited. Only 2 datasets and 2 models are evaluated in this paper. More experiments are needed to verify the effectiveness of proposed method.